# Sex Differences in Coronary Artery Disease and Diabetes Revealed by scRNA-Seq and CITE-Seq of Human CD4+ T Cells

**DOI:** 10.3390/ijms23179875

**Published:** 2022-08-30

**Authors:** Ryosuke Saigusa, Jenifer Vallejo, Rishab Gulati, Sujit Silas Armstrong Suthahar, Vasantika Suryawanshi, Ahmad Alimadadi, Jeffrey Makings, Christopher P. Durant, Antoine Freuchet, Payel Roy, Yanal Ghosheh, William Pandori, Tanyaporn Pattarabanjird, Fabrizio Drago, Angela Taylor, Coleen A. McNamara, Avishai Shemesh, Lewis L. Lanier, Catherine C. Hedrick, Klaus Ley

**Affiliations:** 1La Jolla Institute for Immunology, La Jolla, CA 92037, USA; 2Cardiovascular Research Center, Cardiovascular Division, Department of Medicine, University of Virginia, Charlottesville, VA 22904, USA; 3Parker Institute for Cancer Immunotherapy, University of California, San Francisco, CA 94143, USA; 4Department of Microbiology and Immunology, University of California, San Francisco, CA 94143, USA; 5Immunology Center of Georgia, Augusta University, Augusta, GA 30912, USA; 6Department of Bioengineering, University of California San Diego, San Diego, CA 92093, USA

**Keywords:** coronary artery disease, diabetes, CITE-Seq, scRNA-Seq, PBMC

## Abstract

Despite the decades-old knowledge that males and people with diabetes mellitus (DM) are at increased risk for coronary artery disease (CAD), the reasons for this association are only partially understood. Among the immune cells involved, recent evidence supports a critical role of T cells as drivers and modifiers of CAD. CD4+ T cells are commonly found in atherosclerotic plaques. We aimed to understand the relationship of CAD with sex and DM by single-cell RNA (scRNA-Seq) and antibody sequencing (CITE-Seq) of CD4+ T cells. Peripheral blood mononuclear cells (PBMCs) of 61 men and women who underwent cardiac catheterization were interrogated by scRNA-Seq combined with 49 surface markers (CITE-Seq). CAD severity was quantified using Gensini scores, with scores above 30 considered CAD+ and below 6 considered CAD−. Four pairs of groups were matched for clinical and demographic parameters. To test how sex and DM changed cell proportions and gene expression, we compared matched groups of men and women, as well as diabetic and non-diabetic subjects. We analyzed 41,782 single CD4+ T cell transcriptomes for sex differences in 16 women and 45 men with and without coronary artery disease and with and without DM. We identified 16 clusters in CD4+ T cells. The proportion of cells in CD4+ effector memory cluster 8 (CD4T8, CCR2+ Em) was significantly decreased in CAD+, especially among DM+ participants. This same cluster, CD4T8, was significantly decreased in female participants, along with two other CD4+ T cell clusters. In CD4+ T cells, 31 genes showed significant and coordinated upregulation in both CAD and DM. The DM gene signature was partially additive to the CAD gene signature. We conclude that (1) CAD and DM are clearly reflected in PBMC transcriptomes, and (2) significant differences exist between women and men and (3) between subjects with DM and non-DM.

## 1. Introduction

Atherosclerosis, the pathology underlying cardiovascular disease, is a chronic inflammatory disease of large- and medium-sized arteries. Pre-menopausal women are significantly protected from cardiovascular disease [1]. Later in life, cardiovascular risk in women catches up with that of men [2]. Ultimately, myocardial infarctions and other manifestations of cardiovascular disease become the leading cause of death in women older than 85 years in the United States [3]. The reasons for these sex differences in atherosclerosis and cardiovascular disease [4] are not well understood. Here, we focus on such differences, because cardiovascular disease and its sequelae are a major public health problem [5].

Men and women with type 2 diabetes mellitus (DM) have a 2- to 4-fold elevated risk of death and cardiovascular events compared to the general population [6,7]. Despite the decades-old knowledge that DM is a major risk factor for cardiovascular disease, the reasons for this association are only partially understood. DM accelerates the progression of atherosclerotic lesions and leads to defects in the remodeling of plaques even after cholesterol reduction [8]. Newer classes of drugs approved for glucose lowering in DM reduce cardiovascular disease, but the mechanisms of this benefit do not appear to be explained by glucose lowering [9]. Whereas the effects of glucose lowering on cardiovascular outcomes are unclear, LDL-C lowering consistently reduces cardiovascular risk associated with DM. Both type 1 DM and type 2 DM are associated with greater vascular inflammation [10].

Among inflammatory and immune cell markers, the neutrophil–lymphocyte ratio is an easily obtained inflammatory biomarker that independently predicts cardiovascular risk and all-cause mortality [11]. Among the immune cells involved in cardiovascular disease development, accumulating evidence supports the critical role of T cells as drivers and modifiers of this condition [12]. CD4+ T cells are commonly found in atherosclerotic plaques. Some CD4+ T cells recognize specific epitopes in apolipoprotein B (APOB) [13,14,15].

Among adult humans, sex differences in lymphocyte subsets including T cells have been described for multiple ethnic groups, including Europeans, Asians, and Africans. Females have higher CD4+ T cell counts and higher CD4/CD8 ratios than age-matched males, whereas males have higher CD8+ T cell frequencies [16]. The activity and distribution of CD4+ T cell subsets differs between the sexes. For example, naive CD4+ T cells from human females preferentially produce IFNγ upon stimulation, whereas naive T cells from males produce more IL-17 [17].

To elucidate the impact of cardiovascular disease, DM, and sex on CD4+ T cells in blood, we investigated CD4+ T cells from high-quality frozen peripheral blood mononuclear cells (PBMCs) of 61 men and women with or without DM who underwent cardiac catheterization at the University of Virginia. These subjects are part of the Cardiovascular Assessment Virginia (CAVA) cohort [18]. All cells were interrogated by targeted single-cell RNA sequencing (scRNA-Seq) combined with 49 surface protein markers (CITE-Seq).

## 2. Results

### 2.1. Population and Cells

The 61 CAVA participants initially selected for the current study were aged 42–78 years, mostly non-Hispanic Whites. Matched quartets were selected based on sex, Gensini score, DM status, and statin treatment: 17 diabetic men with or without CAD, 17 non-diabetic men with or without CAD, and 16 women with or without CAD. Nine women had diabetes and seven did not. All these patients were statin-treated. In addition, we studied a group of 10 matched non-diabetic men with or without CAD who were not statin-treated (statistical summary of clinical parameters is in Appendix A). Categorical variables were compared by chi-square test and continuous variables by Mann–Whitney test. PBMC tubes were shipped from the central repository on liquid N2, thawed, and processed according to standard operating procedures, resulting in 91 ± 4% cell viability (Appendix A). To avoid batch effects, all cells were hash-tagged for multiplexing, with four samples run per 250,000-well plate (total of 16 plates). The pooled cells were labeled with 49 titrated oligonucleotide-tagged mAbs (Appendix A). After quality controls and three-stage doublet removal, 140,610 single-cell transcriptomes from 61 WIHS participants were successfully analyzed. Among them, 40,821 CD4+ T cells were identified.

### 2.2. Surface Marker Expression

In combined protein and transcript panel single-cell sequencing, non-specific binding contributes to the antibody signal, in part because Fc block is not complete. Additional background is the consequence of unbound oligonucleotide-tagged antibody remaining in the nanowell that was amplified and sequenced. To account for all sources of background, we established a threshold for each antibody, using ridgeline plots separately for CD4+ T cells, CD8+ T cells, B cells, and monocytes. This yielded thresholds for 34 markers (Appendix A).

### 2.3. CD4+ T Cell Identification

CD4+ T cells were identified as CD19−, CD3+, CD4+, and CD8− cells. CD4+ T cells underwent Louvain clustering. Gates were overlaid to show cluster borders in feature maps (Figure 1A). Initially, 17 clusters were identified in CD4+ T cells, but cluster 9 was identified as likely doublets between CD4+ T cells and myeloid cells, considering some myeloid cell markers such as CD33 positivity. Among the other 16 CD4+ T cell clusters, we identified three naïve clusters, one Thone, one Th2, one Tfh, and one Treg cluster, based on the heatmap of the informative surface markers (Figure 1B). Other clusters were identified as central memory (Cm, CD45RA-CD197+) and effector memory (Em, defined as CD45RA-CD197-; cluster 2: PD1+; cluster 7: cytotoxic, CD56+; cluster 8: CCR2+; cluster 13: PDL1+). Cluster 15 was identified as terminally differentiated effector memory (Emra; CD45RA+ CD197-). Cluster 12 was MMP9+ and cluster 16 was *PDCD1*+ based on mRNA expression. Violin plots for the expression of all surface markers in each cluster are shown in Appendix A.

### 2.4. Transcriptomes

Next, we analyzed the transcriptomes of all single cells (Appendix A). We tested gene expression of each cluster against all other clusters in CD4+ T cells (Figure 2A and Appendix A), using Seurat to report the data as log2 fold-change (logFC). The transcriptomes (Figure 2) confirmed the identity of the CD4+ T cell clusters identified by CITE-Seq (Figure 1) and expanded phenotypic information.

Some CD4+ T cells are cytotoxic [12]. Here, we found significant overexpression of cytotoxic genes, including *GZMB*, *GZMH*, and *NKG7*, as well as *FGFBP2* and *GNLY* in the cytotoxic Em cluster 7. These genes were also expressed in some cells in cluster 2. *KLRF1*, *KLRK1*, *KLRD1*, *CX3CR1*, and *ZNF683* were unique to cluster 7. Treg signature genes including *FOXP3*, *LGLS3*, *TIGIT*, and *CCR3* were upregulated in Treg cluster 17. CD4+ T cells in cluster 8 (CCR2+ at the surface protein level) also expressed higher *CCR2* and *CCR6* transcripts, and also *KLRB1*, *GZMK*, and *RORC* genes, suggesting that this cluster contains Th17 cells. Two naïve clusters (cluster 1 and 14) expressed similar gene patterns, except higher *ACKR3*, *AOC3*, and *CCL2* in cluster 14 than in cluster 1. Tfh (cluster 5) expressed the *CXCR5* gene, the classical chemokine receptor characteristic for Tfh cells [19]. Clusters 1, 6, and 14 expressed low levels of *KLRB1*.

To test how these CD4+ T cell subsets might be related, we conducted a trajectory analysis by Monocle 3 (Figure 2B). This showed that the naïve CD4+ T cells in clusters 1 and 14 split into Th1 cells (cluster 3) and Treg cells (cluster 17). From a downstream node containing cluster 2 (PD1+ Em), central memory (Cm, cluster 4), and some cluster 3 Th1 cells, CCR2+ Em (cluster 8) emerged. Finally, PD1+ Em (cluster 2) gave rise to cytotoxic Em cells (cluster 7).

### 2.5. CD4+ T Cell Subset Abundance

We asked whether, and if so, how, DM, CAD, and sex correlated with the abundance (cell number) of CD4+ T cell subsets expressed as log odds ratios (cell counts are in Appendix A). We compared the log-odds ratios for each of the CD4+ T cell clusters (Figure 3). The proportion of CCR2+ effector memory CD4+ T cells (cluster 8) was significantly (*p* < 0.01) lower in subjects with CAD than in those without CAD. This was mainly driven by a strong decrease in diabetic men with statin treatment (Figure 3A). CCR2 has been reported to be involved in Treg cell recruitment [20], but this role has not been shown in atherosclerosis. Women showed a significantly lower proportion of cluster 8 than men (Figure 3A).

The proportion of *MMP9*+ CD4+ T cells (cluster 12) was significantly higher in subjects with DM, regardless of CAD status (Figure 3B). This cluster was significantly less abundant in women than men. The proportion of *PDL1*+ effector memory CD4+ T cells (cluster 13) was significantly elevated in males compared to females. This was mainly driven by a high abundance of these cells in males with DM and without CAD (Figure 3C). CD4T-C15 (CD123+ Emra) was significantly decreased in women with CAD (Figure 3D). CD4T-C16 (*PDCD1*+) was increased in CAD+ in diabetic men with statin treatment (Figure 3E). Interestingly, cluster 14 (naïve 2) was observed only in non-diabetic men, regardless of CAD status (Figure 4). There was no significant difference in Treg cell (cluster 17) proportions among CAD+ and CAD− and DM+ and DM− participants.

### 2.6. Differentially Expressed Genes by Disease Status

In all CD4+ T cells, several genes, including *TCF7, LTB*, and *GNAI2*, were upregulated in CAD+ subjects (Figure 5A). In DM, *TCF7* and *GNAI2* were also upregulated, in addition to *LAIR2* and *IFITM3*. The known atherosclerosis gene *IL32* [21] was upregulated in DM- subjects compared to DM+ participants (Figure 5B). More systematic investigation of the overlap showed that 31 genes were significantly upregulated in both CAD and DM (see Appendix A for the genes expressed in at least 20% of the higher expressing cell type (pct.1 > 0.2)).

The *TCF7* locus has been reported to be associated with DM, potentially affecting beta cell function [22,23]. Here, we report that *TCF7* is overexpressed in CD4+ T cells of subjects with CAD, which has not been reported before. G-protein alpha-i2, coded by *GNAI2*, is related to chemokine receptor signaling, including CXCR5 signaling, which is thought to induce Hippo/YAP-dependent DM-accelerated atherosclerosis [24]. Local chemoattractants regulate the movement of CD4+ T cells [25]. Specifically, GNAI2 was found to be required for chemokine-mediated arrest of rolling leukocytes [26,27]. Volcano plots of differentially expressed genes (DEGs) for each cluster and dot plots of the common DEGs in CAD and DM are shown in Appendix A, respectively; the underlying data are summarized in Appendix A.

### 2.7. Pathway Analysis

In CD4+ T cells, several pathways such as “Th1” and “Role of NFAT in the Immune Response and T cell receptor signaling” were upregulated in both CAD+ and DM+ subjects (Figure 5A–C). Nuclear factor of activated T cells (NFAT)-centered signaling pathways were enriched in CAD+ subjects, suggesting that NFAT may have a role in atherosclerosis. NFAT is involved in vascular smooth muscle cell phenotypic transition and migration, endothelial cell injury, macrophage-derived foam cell formation, and plaque calcification [28]. Only in CAD, but not in DM, the “Cytotoxic T Lymphocyte-mediated Apoptosis of Target cells” pathway was highly upregulated, and the “T cell exhaustion signaling” pathway was downregulated. In DM, “IL-8 signaling”, “Integrin signaling”, and “Ephrin receptor signaling” pathways were highly and uniquely upregulated. Circulating IL-8 levels are known to be increased in patients with type 2 DM, and IL-8 is associated with worse inflammatory and cardio-metabolic profiles [29].

We also applied pathway analysis to detect systematic sex differences (Figure 5D). Many genes, including *CD52, JUNB, IFITM3*, and *SLC2A3*, were more highly expressed in females than males. This resulted in significant enrichment for the Th2, Th1, IL-7 signaling, and interferon signaling pathways in females. Only two pathways, p53 signaling and apoptosis signaling, were enriched in men.

### 2.8. Random Forest Analysis

To identify the genes with the highest capability to distinguish between disease groups, we used the random forest machine learning (ML) approach. To decrease the covariates, we focused on the participants treated with statins, constituting the majority of our subjects. TCF7 was the most important gene to separate CAD from non-CAD participants (Figure 6A), regardless of DM status (Figure 6B,C). The bottom panels of Figure 6A–C show overlaid ridge plots for the expression of key genes. In DM, *TCF7, S100A9, IL32*, and *KLF2* were overexpressed in CD4+ T cells from subjects with CAD (Figure 6B). In non-DM subjects, *TCF7, CD52*, and *GNAI2* were overexpressed in CAD (Figure 6C). We correlated the gene ranks in DM and non-DM (Figure 6D), which showed that *S100A9* was more important in DM, and *IFITM3, FOSB*, and *CCL5* were more important in non-DM.

Next, we investigated the driver genes for CAD in males and females separately (Figure 6E,F). Interestingly, *TCF7* was important in males, but not in female participants with CAD. *IFITM3* was the most important gene to separate CAD from low-CAD in women, but was ranked 50th in males. This gene encodes interferon-induced transmembrane protein 3. The discrepancies in gene expression in CD4+ T cells between women and men with and without CAD are striking and unexpected. CD4+ T cells from CAD+ men significantly overexpressed *TCF7, LTB*, and *GNAI2* (Figure 6E, lower panels), whereas the top genes in women were *IFITM3, CD3D*, and *CD52* (Figure 6F, lower panels). We correlated the gene ranks in males and females (Figure 6G), which showed that *TCF7*, *LTB*, and *GNAI2* were more important in males, and *IFITM2*, *IFITM3*, and *SORL1* were more important in females. For the genes ranked in the top 50 in both men and women, we also constructed a dot plot of expression (Figure 6H).

### 2.9. Interaction Analysis

CD4+ T cells interact with myeloid cells in the context of antigen presentation [30], co-stimulation, co-inhibition, and for inflammatory receptor–ligand interactions of cytokines and chemokines. We identified classical, nonclassical, and intermediate monocytes as well as dendritic cells. T cell-myeloid interactions are known to be important for antigen presentation. Here, we focus on predicted interactions with *p* < 0.01, with the communication probability coded by color from low (blue) to high (red). All CD4+ T cell clusters from subjects with CAD showed interactions between LGALS9 and CD44 (Figure 7A), and all except cluster 14 showed this interaction in DM+ subjects (Figure 7B). Galectin-9 (coded by *LGALS9* gene) is a crucial regulator of T cell immunity by inducing apoptosis in specific T cell subpopulations associated with autoimmunity and inflammatory disease [31]. About half of the CD4+ T cell clusters showed L-selectin (*SELL*) interaction with PSGL-1 (*SELPLG*) in both CAD+ and DM+ subjects (Figure 7A,B). Multiple studies have provided evidence supporting key roles of *SELP* and *SELPLG* in atherosclerotic lesion formation, thrombosis, and arterial wall changes [32,33]. The *SELL-SELPLG* interaction was stronger in males than in females (Figure 7E,F). There was no male–female difference in *LGALS9–CD44* interaction. Uniquely, only Tregs (cluster 17) showed CTLA4 interaction with CD86 in both males and females. In non-CAD and non-DM subjects (Figure 7C,D and Appendix A), the interaction pattern of CD4+ T cells with myeloid cells was similar, but significant interactions were fewer. The CTLA4 interaction with CD86 in Tregs was preserved.

## 3. Discussion

It is well known that coronary artery disease and its complications including myocardial infarction manifest differently in men and women [34]. Plaque instability in women is more commonly attributed to erosion, whereas rupture is more common in men [35]. However, much less is known about sex differences in immune cells between CAD cases and controls. Here, we show a surprising male-to-female difference in upregulated genes in blood CD4+ T cells. There is a group of genes that are equally important in males and females (ranked in the top 10): *KLF2*, *CD3D*, *CD3E*, *IL32*, *CD48*, and *CD52*. The importance of *CD3D* and *CD3E* suggests that the signaling strength of the TCR is increased in males and females with CAD. *IL32* is a known atherosclerosis-associated gene [25,36]. *KLF2* is involved in Treg development [37], but the function of KLF2 in CD4+ T cells in the development of CAD is unknown. CD48 on T cells enhances TCR signaling through cis interactions with CD2, LAT, and Lck [38,39] and is related to CD4+ T cell activation in other diseases [40,41]. Human and mouse antigen-activated T cells with high expression of CD52 suppressed other T cells [42]. In DM-prone mice, transfer of lymphocyte populations depleted of CD52hi cells resulted in a substantially accelerated onset of DM [42]. Some genes that are important in predicting CAD in females are in the interferon pathway (*IFITM2, IFITM3*). Genes that are important in men but not women are *TCF7* (a transcription factor mainly expressed in naïve T cells), *GNAI2*, and lymphotoxin B (*LTB*).

The large difference in gene expression in CD4+ T cells between men and women was unexpected. Sex hormones are known to interact with the immune system on multiple levels. Autoimmune diseases are more common in women, a phenomenon also partly attributed to sex hormones [43]. Estrogen can be atheroprotective [43]. The expression of several immune response genes in human PBMCs, including *GATA3*, *IFNG*, *IL1B*, *LTA*, *NFKB1*, *PDCD1*, *STAT3*, *STAT5A*, *TBX21*, *TGFB1* and *TNF*, changes during the menstrual cycle [44]. In the present study, we showed that many immune response genes were highly expressed in women, along with many immune response signaling pathways. Furthermore, we showed by random forest analysis that many immune response genes such as *IFITM2, IFITM3*, and *IL32* are more important to separate CAD vs. non-CAD in females than males. Consistent with a more active type I interferon system in women, higher levels of IFNα2 were detected in female COVID-19 patients than in male patients in the same cohort [45]. Conversely, *TCF7, GNAI2, LTB, S100A9, TNFRSF25*, and *CD7* were much more important predictors of CAD in males than females. These genes do not neatly fall into one pathway; further work is needed to identify their function in CD4+ T cells in CAD.

Despite the decades-old knowledge that DM is a major risk factor for cardiovascular disease, the reasons for this association are only partially understood. Interestingly, DM-accelerated atherosclerosis seems to be a human phenomenon and is not reproduced well in mouse models of atherosclerosis [46]. Here, we show a significant overlap in CD4+ T cell gene expression in subjects with CAD and DM, suggesting that the same genes may be important in both pathologies.

scRNA-Seq studies in PBMCs are attractive, because PBMCs (1) are collected in many clinical studies and (2) do not require enzymatic digestion, and (3) data analysis generates a wealth of information. scRNA-Seq has been applied to human PBMCs in atherosclerosis [47,48], and one other study combined single-cell transcriptomes with surface marker expression (CITE-seq [49,50]). The present study is the largest scRNA-Seq and CITE-Seq study in CD4+ T cells. We found significant changes in cell proportions and gene expression patterns in subjects with DM or CAD. As discussed above, some of the key CAD driver genes seem to be sex-specific, but most of the CAD driver genes are similar in subjects with and without DM.

*TCF7* was strongly upregulated in both CAD and DM, especially in men. The association of the *TCF7* locus with T1DM is known [23], but the function of TCF7 has not been reported in CAD. *TCF7* encodes transcription factor 7, also known as T cell-specific transcription factor-1 (TCF-1). TCF7 acts with LEF1 to control the maintenance and functional specification of Treg subsets to prevent autoimmunity [51]. The current study suggests that this gene may connect CAD and DM.

The cell proportions of some clusters, including cluster 8 (CCR2+ Em), 12 (MMP9+), and 13 (PDL1+ Em), were lower in females than males. Among them, CCR2+ Em is a stable population of memory CD4+ T cells equipped for rapid recall response [52]. The role of CCR2 in CD4+ T cells in the context of cardiovascular disease has not been elucidated. 12 Decreased atherosclerotic lesion formation in *CCR2*^−/−^ mice was reported [53], but this is most likely due to limited recruitment of CCR2+ classical monocytes in these mice. In this study, the age of women was 67 ± 8 years (mean ± SD), which is well post-menopausal, but prior to the peak of deaths from cardiovascular disease in women, which is at 85 years old in the U.S. [3].

We showed that among CD4+ T cell interactions with myeloid cells, PSGL-1-L-selectin interactions were seen in more CD4+ T cell subsets in males compared to females. L-selectin on T cells can interact with PSGL-1 on myeloid cells (where it is differentially glycosylated [54]) and thus mediates secondary T cell tethering, which occurs when a freely flowing leukocyte transiently interacts with a rolling or adherent leukocyte or adherent leukocyte fragments and subsequently rolls on the endothelium [55]. Indeed, differential expression or glycosylation of PSGL-1 in different leukocytes mediates selective recruitment of different subsets of monocytes and lymphocytes to atherosclerotic arteries. 58 PSGL-1-expressing cytotoxic CD4+ T cells may induce endothelial cell apoptosis in perimenopausal women with low estradiol levels [56]. Importantly, T cell PSGL-1 is not glycosylated correctly to interact with L-selectin [57]. Thus, T cell L-selectin is most likely to interact with myeloid cell PSGL-1, as shown in Figure 7.

A strength of the present study is that we included well-matched males and females, DM+ and DM− and CAD+ and CAD− participants, obtained a large number of cells, and analyzed surface phenotypes plus transcriptomes. A limitation of the present study is that the number of subjects studied was insufficient to perform multivariate analysis. Cost is a significant barrier to sample size in this type of study; thus, we carefully matched our experimental groups for key clinical variables associated with atherosclerosis to help ameliorate this limitation. Most of the CAVA participants were treated with statins, to the standard of care. Interestingly, all CAD+ subjects not on statins were males. It is known that statin compliance is lower in males than in females. Moreover, most of the participants are non-Hispanic Whites. This hypothesis-generating study awaits a validation cohort. The hypotheses generated may be tested in animal models of atherosclerosis.

## 4. Materials and Methods

### 4.1. Human Subjects

Subjects with suspected coronary artery disease (age range: 40–80 years old) from the Coronary Assessment in Virginia cohort (CAVA) were recruited for the study through the Cardiac Catheterization Laboratory at the University of Virginia Health System, Charlottesville, VA, USA. All participants provided written informed consent before enrollment, and the study was approved by the Human Institutional Review Board (IRB No. 15328). Peripheral blood was obtained from these participants prior to catheterization.

### 4.2. Quantitative Coronary Angiography (QCA)

Patients underwent standard cardiac catheterization with two orthogonal views of the right coronary artery and four of the left coronary artery according to accepted standards. QCA was performed using automatic edge detection from an end-diastolic frame. For each lesion, the frame was selected based on demonstration of the most severe stenosis with minimal foreshortening and branch overlap. The minimum lumen diameter, reference diameter, percent diameter stenosis, and stenosis length were calculated. Analysis was performed by blinded, experienced investigators. The Gensini score [58] was used to assign a score of disease burden to each patient. Briefly, each artery segment is assigned a score of 0–32 based on the percent stenosis. The severity score for each segment was multiplied by 0.5–5, depending on the location of the stenosis. Scores for all segments were then added together to give a final score of angiographic disease burden. Score adjustment for collateral was not performed for this study. Subjects with a Gensini score >30 were classified as high coronary artery disease (CAD) severity subjects and subjects with a Gensini score <6 were classified as low CAD severity subjects.

### 4.3. Preparation of PBMC Samples for CITE-Seq

Peripheral blood from coronary artery disease subjects as well as subjects who had undergone cardiac catheterization to exclude CAD was drawn into BD K2 EDTA vacutainer tubes and processed at room temperature (RT) within one hour of collection. PBMCs were isolated by Ficoll-Paque PLUS (GE Healthcare Biosciences AB, Uppsala, Sweden) gradient centrifugation using SepMate-50 tubes (Stemcell Technologies Inc., Vancouver, BC, Canada) following the manufacturer’s protocol. Trypan blue staining of PBMCs was performed to quantify live cell counts. PBMCs were cryopreserved in freezing solution (90% FBS with 10% DMSO). PBMC vials were stored in Mr. Frosty (Thermo Fisher, Waltham, MA, USA) for 48 h at −80 °C and were then stored in liquid nitrogen until use. To avoid batch effects, 8 samples each were processed on the same day, thawed in a 37 °C water bath, and centrifuged at 400× *g* for 5 min, and pellets were resuspended in cold staining buffer. All reagents, manufacturers, and catalogue numbers are listed in Appendix A. The viability and cell count of each tube were determined using the BD Rhapsody Scanner (Appendix A). The tubes were then centrifuged at 400× *g* for 5 min and resuspended in a cocktail of 51 AbSeq antibodies (2 μL each and 20 μL of SB, listed in Appendix A) on ice for 30–60 min per the manufacturer’s recommendations, then washed and counted again. Of the 65 samples investigated, 61 passed quality control (cell viability > 80%). Cells from each subject were sample tagged using a Sample Multiplexing Kit (BD Biosciences, San Jose, CA, USA) which contains oligonucleotide cell labeling, then washed 3 times, mixed, counted, stained with the 49 antibody mix, washed 3 times again, and loaded onto Rhapsody nanowell plates (4 samples per plate).

### 4.4. Library Preparation

Pre-sequencing quality control (QC) was performed using Agilent TapeStation high-sensitivity D1000 screentape. Each tube was then cleaned with AMPure XP beads and washed in 80% ethanol. The cDNA was eluted, and a second Tapestation QC was performed and diluted as needed. The samples were pooled and sequenced as recommended: AbSeq, 40,000 reads per cell; mRNA, 20,000 reads per cell; sample tags, 600 reads per cell on Illumina NovaSeq using S1 and S2 100 cycle kits (Illumina, San Diego, CA, USA) (67 × 8 × 50 bp). FASTA and FASTQ files were uploaded to the Seven Bridges Genomics pipeline (https://www.sevenbridges.com/apps-and-pipelines/, accessed on 9 November, 2020), where the data were filtered to generate matrices and CSV files. This analysis generated draft transcriptomes and surface phenotypes of 213,515 cells (496 genes, 51 antibodies). After removing multiplets based on sample tags and undetermined cells, 175,628 cells remained. Doublet Finder (https://github.com/chris-mcginnis-ucsf/DoubletFinder, accessed on 7 December, 2020) was used to remove additional doublets, leaving 162,454 cells. Then, 961 CD4+ T cells were removed because they looked like biological doublets with myeloid cells [59]. CD4+ T cells were defined as CD19−, CD14−, CD16−, CD3+, CD4+, and CD8−. In total, 40,821 CD4+ T cells were detected. All antibody data were CLR (centered log-ratio)-normalized and converted to log2 scale. All transcripts were normalized by total UMIs of a gene across all cells, scaled to 1 million, and converted to a log2 scale.

### 4.5. Thresholding

Each antibody threshold (Appendix A) was obtained by determining its signal in a known negative cell or by deconvolution of overlapping normal distributions (we used the function “normalmixEM” to deconvolute the overlapping distributions in the package “mixtools”). To identify the thresholds, ridgeline plots for each antibody in each main cell type were used to set the best threshold (Appendix A). Removing the non-specific binding is critical for correct cell surface phenotypes and correct clustering. It is analogous to isotype controls in flow cytometry.

### 4.6. Clustering

We used UMAP (Uniform Manifold Approximation and Projection) dimensionality reduction in order to project the cells onto the 2D space. We selected UMAP as our chosen algorithm because it focuses on capturing the local similarities while at the same time preserving the global structure of the data. We ran the UMAP algorithm over the first 20 principal components given by Harmony (https://portals.broadinstitute.org/harmony/index.html, accessed on 15 December, 2020). In order to cluster the data, we used the standard Louvain clustering algorithm with the parameter resolution set to 0.15 and random seed set to 42 in order to ensure the reproducibility of the results. Before running the Louvain clustering, we filtered out antibodies that are not expressed (CD19 and CD8 for CD4+ T cells). CD19 was used for the identification of B lymphocytes, CD14 and CD16 for macrophages, CD3 for T lymphocytes, and CD8 for CD8 T lymphocytes.

### 4.7. Trajectory Inference with Monocle 3

Monocle Version v3.10 (http://cole-trapnell-lab.github.io/monocle3/, accessed on 16 November, 2021) was used for trajectory inference in the CD4+ T cell population. Data were first normalized, and the first 20 principal components were computed based on the default PCA method. The cells were projected into a low-dimensional space with the help of Monocle 3’s inbuilt UMAP dimensionality reduction function. Cells were then clustered with the help of the Louvain method with default clustering parameters. Monocle 3 uses a principal graph embedding algorithm to learn cell trajectories in a low-dimensional space. To construct trajectories, the root node was anchored at naïve CD4+ T cells (clusters 1 and 14) and the pseudo-time was estimated.

### 4.8. Comparing Cell Proportions

To find changes in proportions, we identified the cell numbers for each participant in each cluster, and calculated the log-odds ratio *p*/(1 − *p*), where *p* is the proportion of cells, followed by pairwise comparison.

### 4.9. Comparing Gene Expression among Participant Types

To determine differential expression (DE), we used the Seurat package in R (https://satijalab.org/seurat/, accessed on 21 August, 2021) with thresholds set to 0 for average log2 fold change, the minimum fraction of cells required in the two populations being compared, the minimum number of cells in one of the groups, and the minimum number of cells expressing a feature as per Seurat defaults in either group, and further filtered for adjusted *p*-value < 0.05.

### 4.10. Random Forest Model

A machine learning (ML) approach was implemented to identify the genes with the highest capability to distinguish between disease groups. To accomplish this goal, the random forest (RF) model [60,61] was trained with the normalized gene expression from 1000 randomly selected cells from each condition, and variable importance scores of the genes were calculated. This procedure was repeated for 15 iterations and importance scores in each iteration were scaled to the 0–100 range for a better comparison. A higher score means that the gene is more important for classifying the groups.

### 4.11. Cell interaction Analysis

CellChat Version v1.1.3 (https://github.com/sqjin/CellChat, accessed on 3 November, 2021) was used to infer and compare intracellular communication between cell types and conditions. A CellChat object was created with normalized data cells grouped by their cluster labels to infer cell communication between clusters. The CellChat-curated human ligand receptor database was used to validate the molecular interactions in the dataset. CellChat computes probabilities for biologically significant communication patterns by assessing and integrating gene expression levels along with prior known knowledge for molecular interactions. Bubble plots were created based on the communication probabilities computed for ligand receptor pairs by CellChat’s algorithm.

## 5. Conclusions

This study of CD4+ T cells provides evidence for strong sex differences in the gene expression and abundance of CD4+ T cell subtypes. We conclude that (1) CAD and DM are clearly reflected in PBMC transcriptomes, and (2) significant differences exist between women and men and (3) between subjects with DM and non-DM. Gene expression patterns and pathways associated with CAD and DM are overlapping, suggesting shared mechanisms.

## Figures and Tables

**Figure 1 ijms-23-09875-f001:**
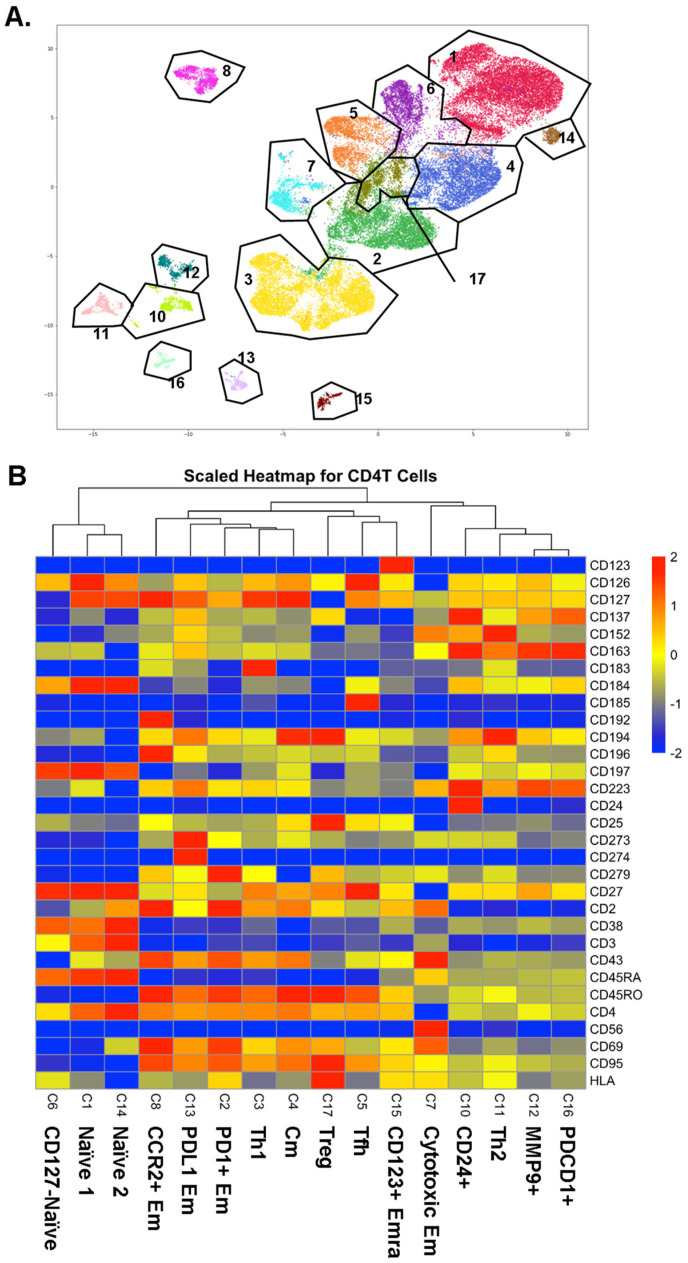
(**A**) CD4+ T cells were UMAP–Louvain-clustered by all non-negative surface markers. CD4+ T cells formed 16 clusters; gates (black lines) overlaid and cluster numbers indicated. (**B**) Scaled (log2) heatmaps of antibody expression in each CD4+ T cell cluster. Cluster numbers correspond to panel A; cluster names indicated. Mean expression: yellow, highest: red, lowest: blue.

**Figure 2 ijms-23-09875-f002:**
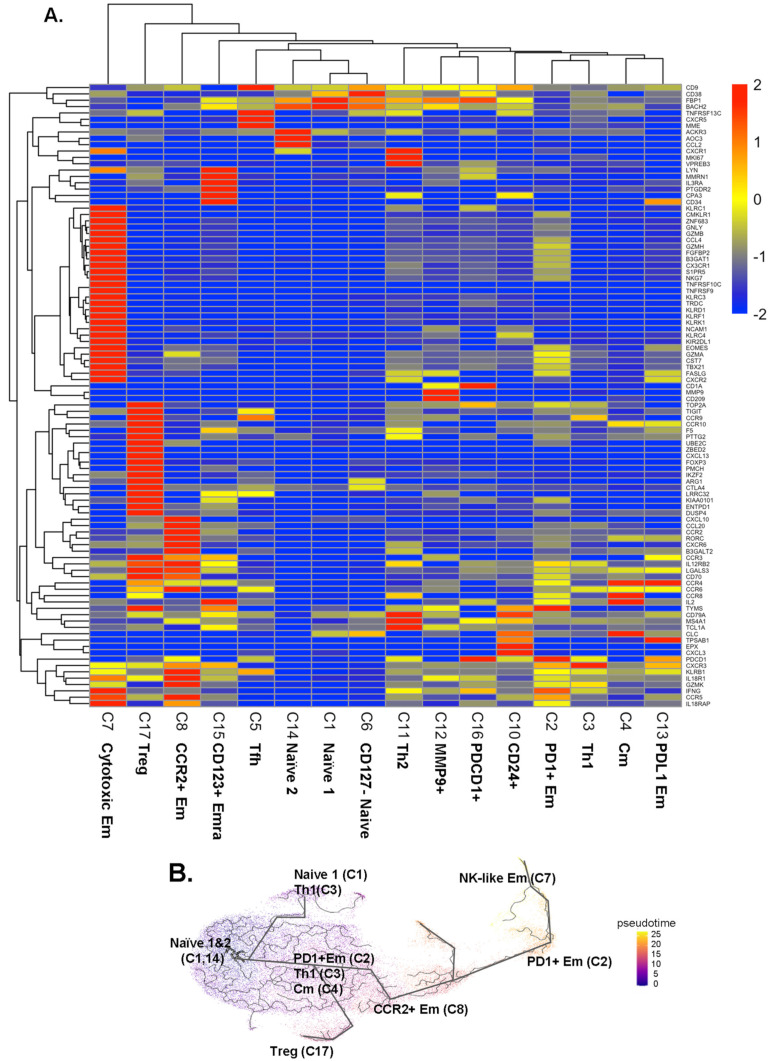
(**A**) Scaled heatmaps of differentially expressed genes (filtered on the basis of adjusted *p*-value < 0.05, avg_log2FC > 0, and pct.1 (percent of cells expressing each gene in each cluster)/pct.2 (percent of cells expressing each gene in all other clusters) >2.5 in each subcluster. (**B**) Trajectory analysis by Monocle 3. The starting node was defined as naïve clusters 1 and 14. Pseudo-time indicated by color (blue to yellow). Mean expression: yellow, highest: red, lowest: blue.

**Figure 3 ijms-23-09875-f003:**
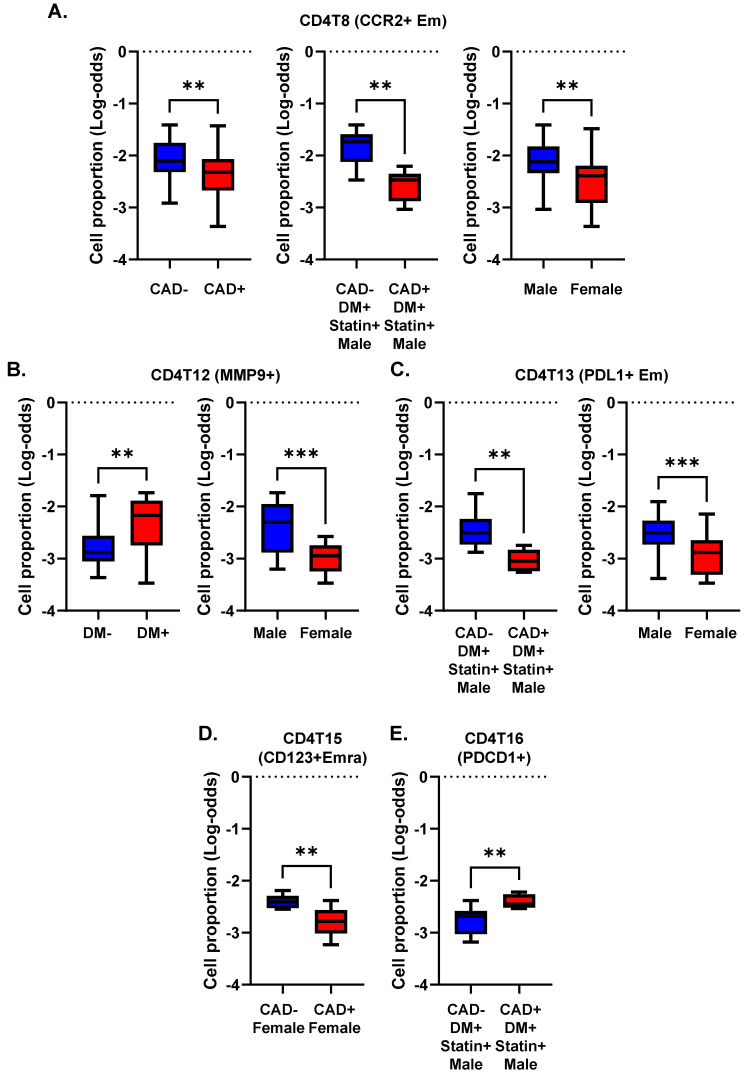
Cell proportions in men and women with or without CAD and/or DM. (**A**–**E**) Proportions of cells in each cluster calculated as percentage of the parent cell type as indicated in the title of each panel. Clusters with significant differences (** *p* < 0.01, *** *p* < 0.001) in cell proportions (by log odds ratio) are shown with means and standard error of the mean (SEM).

**Figure 4 ijms-23-09875-f004:**
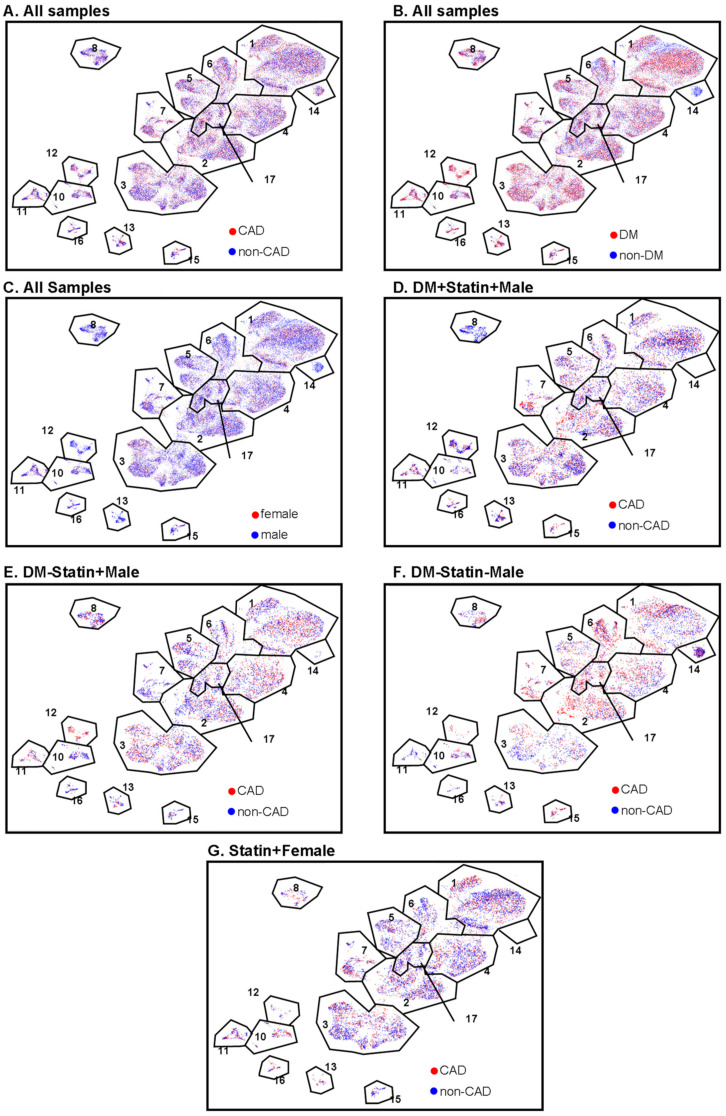
CAD, DM, and sex features projected onto CD4+ T cell UMAPs. (**A**) CAD (red) and non-CAD (blue) projected to UMAPs from all the samples. (**B**) DM (red) and non-DM (blue) projected to UMAP from all the samples. (**C**) Female (red) and male (blue) projected to UMAP from all the samples. (**D**–**G**) CAD (red) and non-CAD (blue) projected to UMAP from DM + statin + male samples (**D**), DM-statin + male samples (**E**), DM-statin-male (**F**), and all statin + females (**G**). Gates (black polygons) and cluster numbers from Figure 1.

**Figure 5 ijms-23-09875-f005:**
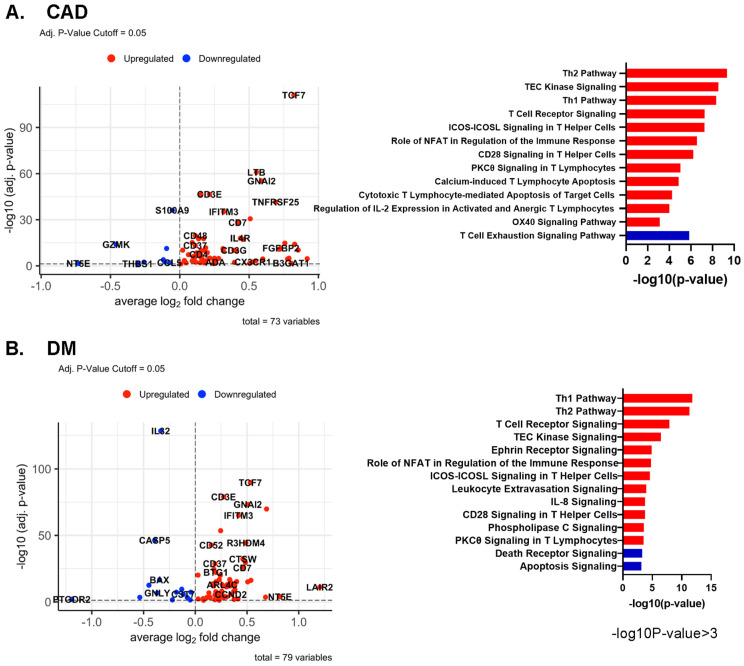
(**A**,**B**) Volcano plots comparing gene expression in single cells and Ingenuity Pathway analysis (−log10 *p*-value > 3) of CAD high vs. CAD low (**A**) and of DM+ vs. DM− (**B**) in all CD4+ T cells. Red bar means positive Z-score, and blue bar means negative Z-score. (**C**) Dot plots of differentially expressed genes between CAD+ vs. CAD−, and between DM+ vs. DM− in all CD4+ T cells. The thresholds set for the plots were adjusted *p*-value < 0.05, avg.Log2FC > 0 or <0, and pct.1 > 0.2. The size of dots represents log(pct.1/pct.2), where pct.1 is the proportion of cells expressing each gene in DM or CAD+ and pct.2 is the proportion of cells expressing each gene in DM− or CAD−. (**D**) Volcano plots comparing gene expression in single cells and Ingenuity Pathway analysis (−log10 *p*-value > 3) for gene expression in CD4+ T cells from male and female subjects. Red bar means positive Z-score, blue bar means negative Z-score.

**Figure 6 ijms-23-09875-f006:**
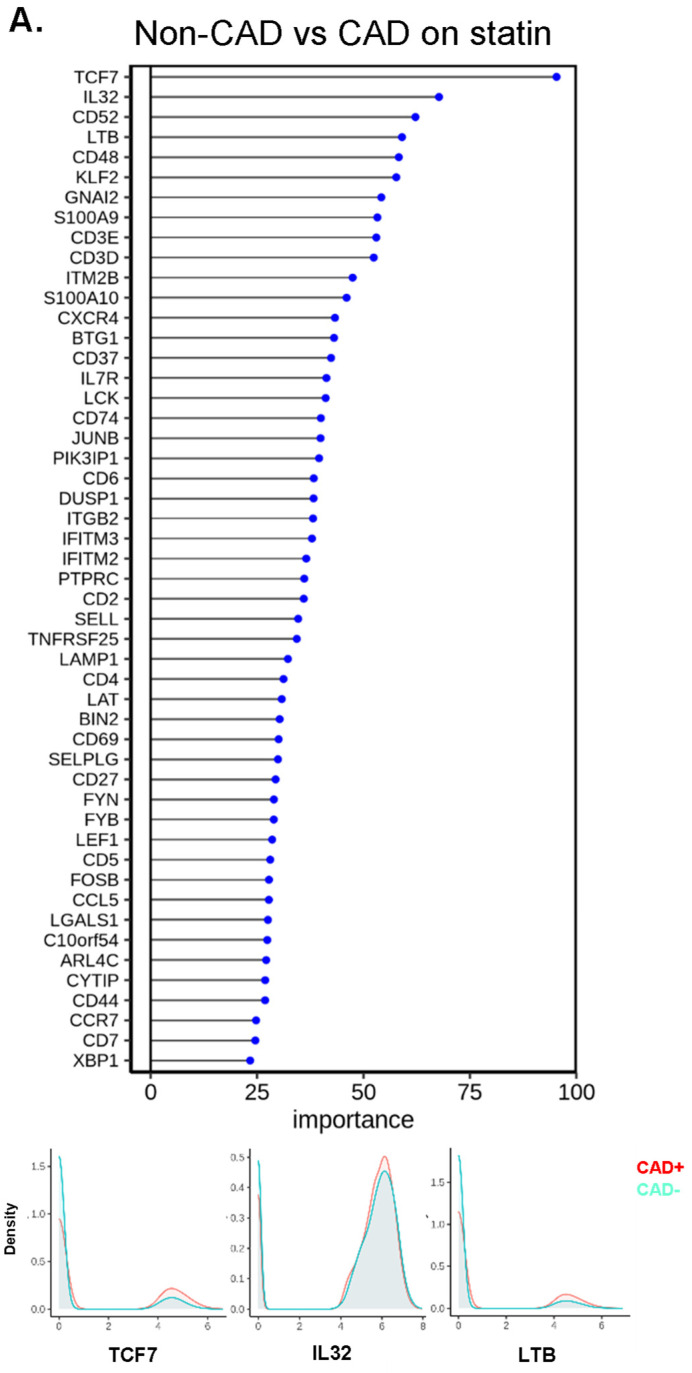
Machine learning analysis using the random forest model. (**A**) Line with dot plots which show the feature importance of each gene in the comparison of non-CAD vs. CAD. Below, representative density plots are shown. (**B**,**C**,**E**,**F**) Line with dot plots for the feature importance of each gene in the comparison of non-CAD vs. CAD in DM (**B**), in non-DM (**C**), in males (**E**), and in females (**F**). At the bottom of each plot, representative density plots are shown. (**D**,**G**) Correlation of the ranks of the important genes for CAD in DM (*x*-axis) vs. non-DM (*y*-axis), and males (*x*-axis) and females (*y*-axis). Red dot, rank in DM < rank in non-DM; blue, rank in DM > rank in non-DM; yellow, rank in DM = rank in non-DM in (**D**). Red dot, rank in male < rank in female; blue, rank in male > rank in female; yellow, rank in male = rank in female in (**G**). When a gene ranked outside 50, the actual ranks are shown (x,y). (**H**) Dot plot showing the importance of genes for CAD in males and females side by side. Average log2 fold change (log2FC) for male to female comparison (red, higher in males; blue, higher in females). Genes were sorted based on the importance in females.

**Figure 7 ijms-23-09875-f007:**
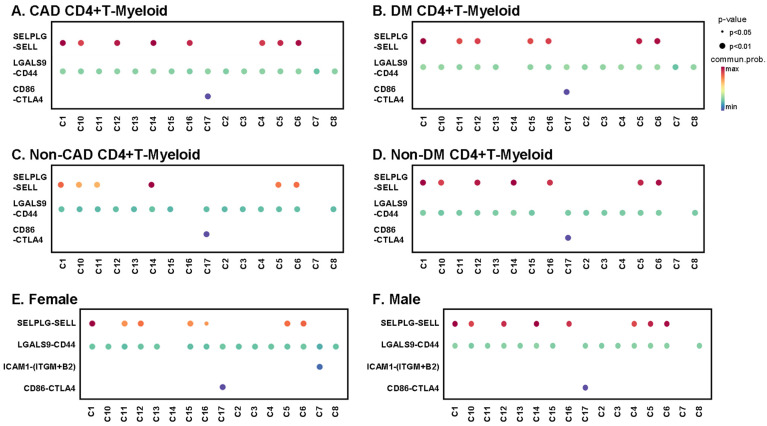
Bubble plots of interaction analysis between each CD4+ T cell cluster and myeloid cells. The color bar indicates communication probability from lowest (blue) to highest (red). The size of bubbles indicates *p*-value. Interactions between CD4+ T cells and myeloid cells in CAD+ subjects (**A**), DM+ subjects (**B**), non-CAD subjects (**C**), non-DM subjects (**D**), females (**E**) and males (**F**). More interactions in Appendix A.

## Data Availability

All data are available at GEO: GEO access number is GSE190570.

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
