# Peer review of "Sex Differences in Coronary Artery Disease and Diabetes Revealed by scRNA-Seq and CITE-Seq of Human CD4+ T Cells"

_ijms, 2022, doi:10.3390/ijms23179875_

Round 1

Reviewer 1 Report

This study investigated  the relation of CAD with sex and diabetes mellitus  by single-cell RNA and antibody sequencing  of CD4+ T cells. The  findings showed  strong sex differences in the gene  expression and abundance of CD4+ T cell subtypes, and overlapping gene expression patterns and pathways associated with CAD and DM.

The present manuscript deals with a subject of interest, and the findings are believable.   

However, there are several limitations that Authors should address

1- The authors should clarify the time of collection of blood samples: before or after the cardiac catheterization?

2-The Gensini scoring system is used for  quantifying the severity, not the presence of  CAD. Please clarify.

3.-Overall, the paper is quite challenging to follow in its present form . The manuscript is too “full” of data and there are some problems with clarity of message. Thus, the paper would benefit from a careful extensive rewriting to improve the clarity of the article (e.g. general aim and findings).

Author Response

Reviewer 1

This study investigated  the relation of CAD with sex and diabetes mellitus  by single-cell RNA and antibody sequencing  of CD4+ T cells. The  findings showed  strong sex differences in the gene  expression and abundance of CD4+ T cell subtypes, and overlapping gene expression patterns and pathways associated with CAD and DM.

The present manuscript deals with a subject of interest, and the findings are believable.   

However, there are several limitations that Authors should address

  • The authors should clarify the time of collection of blood samples: before or after the cardiac catheterization?

The blood samples were collected before the cardiac catheterization. We wrote it in the method section (2.1 Human subjects. ‘Peripheral blood was obtained from these participants prior to catheterization.’)

  • The Gensini scoring system is used for  quantifying the severity, not the presence of  CAD. Please clarify.

Thank you for pointing it out. We now revised the abstract to read ‘CAD severity was quantified using Gensini scores, with scores above 30 considered CAD+ and below 6 considered CAD-‘.

3.-Overall, the paper is quite challenging to follow in its present form . The manuscript is too “full” of data and there are some problems with clarity of message. Thus, the paper would benefit from a careful extensive rewriting to improve the clarity of the article (e.g. general aim and findings).

We added a new conclusions section at the end of the discussion.

Reviewer 2 Report

This manuscript, written by Saigusa-san, made a comparison of the characteristics of the CD4+T cells in the blood of coronary artery. The properties of the CD4+cells were established by the analysis of surface markers (that made the clusters) and the gene expression. Then, CAD+ vs -, DM+ vs DM-, and male vs. female were compared.

 The manuscript is well written. The research uses state of the art technology. There are many variables and statistical comparisons; this difficulty has been acknowledged by the authors.

 (A) I was not sure if there was a group of women with diabetes.

(B) I wonder why the characteristics of the CD4+Tcells are supposed to be so related to the coronary disease. Why not studying the tissue of the arteriosclerotic plaque?

 Specific comments:

 (1) Lines 46-53. Could you please mention the protective effect of estrogens?

 (2) Lines 73-73. “Females have higher CD4+ T cell counts and higher CD4/CD8 ratios than age-matched males”. Could the higher frequency of CD4+T helper cells in women explain the higher incidence of autoimmune diseases?

 (3) Lines 76-76. “For example, naive CD4+ T cells from human females preferentially produce IFNgamma upon stimulation, whereas naive T cells from males produce more IL-17”. Is this difference present in all parts of the body? Or is it tissue specific (for example, intestine)?

 (4) Lines 79-81. “investigated CD4+ T cells from high-quality frozen peripheral blood mononuclear cells (PBMCs) of 61 men and women with or without DM who underwent cardiac catheterization at the University of Virginia”. What kind of treatment were the patients receiving during the peripheral blood extraction? Could the ongoing treatment affect the expression of T cells? Could the process of thawing frozen blood change the surface expression of the target cells?

 (5) Regarding the Genisi score. Could you please add the score as a table in the appendix or the supplementary files?

 (6) Line 108. “Blood from coronary artery disease”. I think this is crucial and should be clearly stated in the abstract. Additionally, how different is this coronary artery blood different from other peripheral blood?

 (7) Regarding section 2.3. I understand that the cells were labeled with oligo-antibodies. Then, the cells were loaded onto cartridge and the cell capture beads were retrieved. For the capture, are you using CD4 antigen? Why are the rest of 48 antibody needed for? (page 126).

 (8) Lines 141-142. Could you please write that CD19 is for the identification of B lymphocytes, CD14 and CD16 for macrophages, CD3 for T lymphocytes, and CD8 for cytotoxic T lymphocytes?

 (9) Line 135. “Seven Bridges Genomics pipeline”. Could you please provide more information about the software that was used? Was it the aria, graf, rheo, or the platform?

 (10) Lines 143-145, which genes were used for the housekeeping gene normalization?

(11) Line 146. Could you please explain why the antibody threshold is necessary? (Of note, later it is explained)

 (12) Line 157. Could you please add the link for harmony? (https://portals.broadinstitute.org/harmony/index.html) (bioRxiv 2019. doi.org/10.1101/461954).

 (13) Line 177.  Regarding Seurat.

The following links could be added: https://satijalab.org/seurat/

If you use Seurat in your research, please consider citing:

    Hao*, Hao*, et al., Cell 2021 [Seurat V4]

    Stuart*, Butler*, et al., Cell 2019 [Seurat V3]

    Butler* et al., Nat Biotechnol 2018 [Seurat V2]

    Satija*, Farrell*, et al., Nat Biotechnol 2015 [Seurat V1]

An alternative: https://cloud.r-project.org/web/packages/Seurat/index.html

 (14) Line 204-205. “17 diabetic men with or without CAD, 17 non-diabetic men with or without CAD, and 16 women with or without CAD.”. Could you please explain why there are no diabetic women?

 (15) In Figure 1. The CD4+cells are clustered using only the surface markers, aren’t they?

 (16) In Figure 2. The clusters the same as in Figure 1?

 (17) Regarding cluster 12, what type of Th cell is MMP9 positive?

 (18) Regarding Figure 3, line 293. “Cell proportions in men and women with or without CAD and/or DM”. Do you have women with diabetes?

(19) Any information regading PTX3?

(20) Any information regarding Th17?

(21) Between DM+ vs. DM-, do Tregs differ?

(22) It is long manuscript and complex. Please try to simplify as much as possible, or move less important parts to appendix or supplementary.

Author Response

Reviewer 2

This manuscript, written by Saigusa-san, made a comparison of the characteristics of the CD4+T cells in the blood of coronary artery. The properties of the CD4+cells were established by the analysis of surface markers (that made the clusters) and the gene expression. Then, CAD+ vs -, DM+ vs DM-, and male vs. female were compared.

 The manuscript is well written. The research uses state of the art technology. There are many variables and statistical comparisons; this difficulty has been acknowledged by the authors.

  • I was not sure if there was a group of women with diabetes.

In this study, 9 of 16 women had diabetes (please see Table S5B). The women were included in the diabetes analysis (Figure 5C and Figure S4).

  • I wonder why the characteristics of the CD4+Tcells are supposed to be so related to the coronary disease. Why not studying the tissue of the arteriosclerotic plaque?

Others have studied immune cells in atherosclerotic plaque (see refs #51-52). In this study, we asked whether CAD leaves an imprint on CD4 T cell transcriptomes. This is based on the knowledge that CD4 T cells are important in cardiovascular disease (see refs. #12-15). Indeed, we found significant differences. This is important, because PMBCs are more easily available in clinical practice (simple blood draw).

 Specific comments:

  • Lines 46-53. Could you please mention the protective effect of estrogens?

We added the sentence in the introduction.

‘Estrogen can be atheroprotective.47’

 (2) Lines 73-73. “Females have higher CD4+ T cell counts and higher CD4/CD8 ratios than age-matched males”. Could the higher frequency of CD4+T helper cells in women explain the higher incidence of autoimmune diseases?

There is no evidence that the higher frequency of CD4T cells explains the higher incidence of autoimmune diseases in females.

 (3) Lines 76-76. “For example, naive CD4+ T cells from human females preferentially produce IFNgamma upon stimulation, whereas naive T cells from males produce more IL-17”. Is this difference present in all parts of the body? Or is it tissue specific (for example, intestine)?

A previous study (ref # 17) reported the difference in cytokine production in PBMCs. To the best of our knowledge, there was no report regarding other human tissues.

 (4) Lines 79-81. “investigated CD4+ T cells from high-quality frozen peripheral blood mononuclear cells (PBMCs) of 61 men and women with or without DM who underwent cardiac catheterization at the University of Virginia”. What kind of treatment were the patients receiving during the peripheral blood extraction? Could the ongoing treatment affect the expression of T cells? Could the process of thawing frozen blood change the surface expression of the target cells?

Some of the patients were receiving diuretics, beta blockers, calcium channel blockers, ACE inhibitors, ATRs, NSAIDs, all of which are listed in Table S5. Of course, these medications can affect the transcriptomes of CD4 T cells. Thawing frozen PBMCs can introduce some artifacts, like B-T cell aggregates (Elife 2019. Circulating T cell-monocyte complexes are markers of immune perturbations). We eliminated such aggregates and they were not analyzed.

 (5) Regarding the Genisi score. Could you please add the score as a table in the appendix or the supplementary files?

The Gensini scores are it in Table S5 in the supplement.

 (6) Line 108. “Blood from coronary artery disease”. I think this is crucial and should be clearly stated in the abstract. Additionally, how different is this coronary artery blood different from other peripheral blood?

This sentence was misleading. It was corrected to read “peripheral blood from subjects with coronary artery diseases” (methods section page 3).

 (7) Regarding section 2.3. I understand that the cells were labeled with oligo-antibodies. Then, the cells were loaded onto cartridge and the cell capture beads were retrieved. For the capture, are you using CD4 antigen? Why are the rest of 48 antibody needed for? (page 126).

CD4 antigen is not needed for the capture. The antibodies were used to study the cell surface phenotype of the cells by CITE-Seq and the transcriptomes by scRNA-Seq (multi-omics approach).

 (8) Lines 141-142. Could you please write that CD19 is for the identification of B lymphocytes, CD14 and CD16 for macrophages, CD3 for T lymphocytes, and CD8 for cytotoxic T lymphocytes?

We added the sentence in section 2,6.

 (9) Line 135. “Seven Bridges Genomics pipeline”. Could you please provide more information about the software that was used? Was it the aria, graf, rheo, or the platform?

Seven Bridges is a platform provided b Becton Dickinson, the manufacturer of the Rhapsody system. We added the link to Seven Bridges in the section 2.4 library preparation.

 (10) Lines 143-145, which genes were used for the housekeeping gene normalization?

We didn’t use housekeeping genes for normalization.

(11) Line 146. Could you please explain why the antibody threshold is necessary? (Of note, later it is explained)

Removing the non-specific binding is critical for correct cell surface phenotypes and correct clustering. It is analogous to isotype controls in flow cytometry. This sentence was added on page 4.

 (12) Line 157. Could you please add the link for harmony? (https://portals.broadinstitute.org/harmony/index.html) (bioRxiv 2019. doi.org/10.1101/461954).

We added the link.

 (13) Line 177.  Regarding Seurat.

The following links could be added: https://satijalab.org/seurat/

If you use Seurat in your research, please consider citing:

    Hao*, Hao*, et al., Cell 2021 [Seurat V4]

    Stuart*, Butler*, et al., Cell 2019 [Seurat V3]

    Butler* et al., Nat Biotechnol 2018 [Seurat V2]

    Satija*, Farrell*, et al., Nat Biotechnol 2015 [Seurat V1]

An alternative: https://cloud.r-project.org/web/packages/Seurat/index.html

We added the link on page 4.

 (14) Line 204-205. “17 diabetic men with or without CAD, 17 non-diabetic men with or without CAD, and 16 women with or without CAD.”. Could you please explain why there are no diabetic women?

9 of the 16 women were diabetic (Table S5B). The women were included in the diabetes analysis (Figure 5C and Figure S4).

 (15) In Figure 1. The CD4+cells are clustered using only the surface markers, aren’t they?

Yes, they are.

 (16) In Figure 2. The clusters the same as in Figure 1?

Yes, they are.

 (17) Regarding cluster 12, what type of Th cell is MMP9 positive?

We couldn’t decide what Th type cluster 12 was from the surface markers. CD9+ CD4 T cells have not been described before. Based on the transcriptome, we called them MMP9+ CD4 T cells.

 (18) Regarding Figure 3, line 293. “Cell proportions in men and women with or without CAD and/or DM”. Do you have women with diabetes?

Yes. We have 4 diabetic women in CAD+ women (n=7), and 5 diabetic women in CAD- women (n=9).

(19) Any information regading PTX3?

PTX3 did not show any difference.

(20) Any information regarding Th17?

Although we did not identify a clear Th17 cluster in this study, we think cluster 8 may contain Th17 cells, judging from CCR6 surface marker expression and RORC gene expression.

(21) Between DM+ vs. DM-, do Tregs differ?

In this study, there was 3 significant gene expression differences in Tregs (cluster 17). Tregs showed higher expression of R3HDM4 and IL-32 in subjects with diabetes and higher IFITM3 in subjects with CAD (Figure S4).

(22) It is long manuscript and complex. Please try to simplify as much as possible, or move less important parts to appendix or supplementary.

We edited the manuscript.

Round 2

Reviewer 1 Report

None